# Academic stress and its psychosocial and behavioral determinants in medical students: Findings from a cross-sectional study

Md Rizwanul Karim[1]*, S. A. Sazin Haque[2], Faiza Rumeen[3], Purna Aruneema[4]

1 Associate Professor (Epidemiology) and Head, Department of Community Medicine and Public Health, Shaheed M. Monsur Ali Medical College (SMMAMC), Sirajganj, Bangladesh, 2 Research Assistant, Department of Medicine, Patuakhali Medical College, Patuakhali, Bangladesh, 3 Research Assistant, Department of Medicine, Patuakhali Medical College, Patuakhali, Bangladesh, 4 Research Assistant, Department of Medicine, Patuakhali Medical College, Patuakhali, Bangladesh

* Shameem.m25@gmail.com

## Abstract

### Background

Academic stress is a widespread challenge in medical education, with psychological, behavioral, and contextual factors contributing to it. This study estimated the prevalence of academic stress among Bangladeshi medical students and identified key psychosocial and behavioral predictors to guide targeted interventions.

### Methods

A multicenter cross-sectional study (October–December 2022) used a stratified random sample of 1,072 undergraduate students from eight public medical colleges representing all administrative divisions of Bangladesh. Validated instruments measured academic stress (Academic Stress Scale, ASS-40), depressive symptoms (PHQ-9), anxiety (GAD-7), insomnia (ISI), internet addiction (IAT), self-esteem (RSES), and coping styles (SCSI). Analyses included descriptive statistics, chi-square and Mann–Whitney U tests, multivariable logistic regression to identify independent predictors, and structural equation modeling (SEM) and network analysis to explore direct and indirect pathways.

### Result

Academic stress was reported by 47.5% of participants. In adjusted logistic regression models, moderate anxiety was associated with increased odds of academic stress (AOR = 3.95; 95% CI 1.98–7.90), and severe depression showed a markedly elevated association (AOR = 21.54; 95% CI 7.21–64.38). Behavioral factors were also influential: moderate-to-severe problematic internet use was strongly associated with academic stress (AOR = 17.78; 95% CI 9.66–32.72). Additional independent

**Data availability statement:** The data that support the findings of this study are openly available in figshare at Karim, Md Rizwanul (2025). Psychological, Behavioral, and Socioeconomic Correlates of Stress in Bangladeshi Medical Students: A Cross-Sectional Study. figshare. Dataset. https://doi.org/10.6084/m9.figshare.29014094.v1.

**Funding:** The author(s) received no specific funding for this work.

**Competing interests:** The authors have declared that no competing interests exist.

predictors included advanced academic year, higher monthly expenditure, and urban residence. Active problem-focused coping conferred modest protection against academic stress (AOR = 0.89; 95% CI 0.83–0.95). Structural equation modeling supported a model in which psychological distress exerted both direct effects on academic stress and indirect effects mediated by sleep disturbance and internet addiction, while network analysis identified depressive symptoms, insomnia, and internet addiction as central nodes within the stress network.

## Conclusions

Nearly half of the sampled medical students experienced significant perceived academic stress. Interventions that integrate mental health services, sleep-hygiene promotion, responsible digital-use policies, and training in adaptive, problem-focused coping are recommended.

## Introduction

Academic stress is a pervasive challenge among medical students worldwide, arising from the interplay of psychological, behavioral, and sociodemographic factors. The demanding nature of medical education—marked by intensive study schedules, high expectations, and frequent high-stakes assessments—places students under considerable psychological strain. Persistent stress undermines academic performance, mental health, and physical well-being, often manifesting as burnout, depression, or anxiety [1,2]. Medical students are particularly vulnerable due to the competitive learning environment and the constant transition between theoretical coursework and clinical responsibilities [3]. Growing recognition of these challenges has stimulated extensive research into the prevalence, determinants, and consequences of academic stress in medical education [4,5].

Psychological determinants are consistently highlighted as central to stress experiences. Depression and anxiety are highly prevalent among medical students and exacerbate stress by impairing emotional regulation, concentration, and cognitive functioning [6–9]. Sleep disturbances, including insomnia and poor sleep quality, are also widespread and linked to psychological distress and reduced academic performance [10]. Beyond psychological vulnerabilities, behavioral factors such as internet addiction and coping strategies significantly shape stress outcomes. Excessive internet use disrupts time management and sleep, thereby intensifying stress [11,12]. Coping strategies further influence resilience: maladaptive mechanisms heighten stress, whereas adaptive, problem-focused approaches mitigate its impact [13,14].

Sociodemographic characteristics also play a role. Studies indicate that parental education, religion, place of residence, and living arrangements affect stress levels and coping capacity [4,14]. For example, students from urban environments often report higher stress than rural peers, potentially due to greater competition and lifestyle pressures [14,15]. Maternal education may influence stress through socioeconomic resources, academic expectations, and emotional support [15]. Living

arrangements likewise shape support networks and routines, with differences observed between students residing with family and those living independently or with peers [16].

Although academic stress has been widely studied internationally, evidence from Bangladesh remains limited. Available studies nonetheless underscore its significance. A cross-sectional study among final-year health science students reported that 68.6% experienced stress symptoms during their final training stage [17]. A multicenter study across Bangladeshi medical colleges found a stress prevalence of 54%, with academic demands identified as the primary stressor [18]. Further research has linked academic frustration and curricular changes to poorer mental health outcomes among private university students [19]. Other investigations have identified personal inadequacy, workload, and inadequate study facilities as major stress-inducing domains among tertiary students [20]. Collectively, these findings confirm that academic stress is widespread in Bangladesh and shaped by both psychological vulnerabilities and contextual academic challenges.

Additionally, the COVID-19 pandemic led to substantial disruptions in Bangladesh's medical education, which in turn significantly affected students' psychological and behavioral experiences, contributing to increased academic stress. Despite this growing body of work, important gaps remain. Few Bangladeshi studies have examined academic stress comprehensively; most addressed isolated domains. As stress is multidimensional, inadequate integration of psychological, behavioral, and sociodemographic factors limits holistic understanding, particularly in resource-constrained contexts such as Bangladesh.

Addressing these gaps has both methodological and societal relevance. Methodologically, examining multiple predictors within a single analytical framework allows a more rigorous assessment of the relative and combined contributions of psychological, behavioral, and sociodemographic factors. Given the multifactorial nature of academic stress, studies that analyze several determinants simultaneously provide more reliable insight into the mechanisms underlying stress and related mental health outcomes while accounting for potential confounding influences [2,21]. From a societal and educational perspective, high levels of stress among medical students are associated with poorer mental health, reduced academic performance, and increased risk of burnout, potentially affecting the quality of the future healthcare workforce [2,22]. Identifying key psychological, behavioral, and contextual determinants can therefore inform institutional strategies to support student well-being, including accessible mental health services, targeted counselling, and structured stress-management programs [23]. In addition, addressing modifiable behavioral factors—such as maladaptive coping patterns and excessive internet use—may facilitate preventive interventions that enhance resilience and promote healthier academic environments.

The objective of this study was to estimate the prevalence of academic stress among Bangladeshi medical students and to identify its principal determinants. Specifically, it examined the associations of academic stress with psychological factors (anxiety and depression), behavioral factors (insomnia, internet addiction, and coping strategies), and selected sociodemographic characteristics (parental education, religion, and living arrangements). By integrating these multidimensional domains into a single analytical framework, the study aimed to provide a holistic understanding of the drivers of academic stress in Bangladeshi medical education and to generate evidence to inform institutional policies and targeted interventions to support student mental health and academic performance.

## Theoretical framework

The proposed conceptual framework is grounded in the Transactional Model of Stress and Coping developed by Richard S. Lazarus and Susan Folkman, which conceptualizes stress as arising from cognitive appraisal processes and coping responses to environmental demands [24]. Guided by this theoretical perspective, psychological distress is conceptualized as the primary explanatory construct underlying academic stress among medical students.

Psychological distress is operationalized as a latent construct indicated by anxiety and depressive symptoms, reflecting their strong comorbidity and shared internalizing dimension in student populations [2,25,26]. Academic stress is specified as the primary outcome variable. A direct positive association between psychological distress and academic stress

is hypothesized, consistent with prior evidence linking anxiety and depression to heightened perceived academic burden and impaired academic functioning [25,27].

Four behavioral variables—sleep quality, self-esteem, internet addiction, and coping style—are theorized to function as mediators in this relationship. Psychological distress has been consistently associated with sleep disturbance [28], lower self-esteem [29], problematic internet use [30], and maladaptive coping strategies [31]. Each of these factors, in turn, has been independently linked to greater academic stress and poorer academic adjustment [32–34]. Accordingly, the framework posits that psychological distress influences academic stress both directly and indirectly through these behavioral pathways [Fig 1].

Sociodemographic characteristics (sex, living place, family type, maternal education, family expenditure, and academic year) are incorporated as exogenous control variables, based on evidence that contextual and socioeconomic factors shape mental health vulnerability and stress experiences in university populations [35–37]. These variables are included to adjust for potential confounding effects within the structural model. This theory-driven framework will be tested using

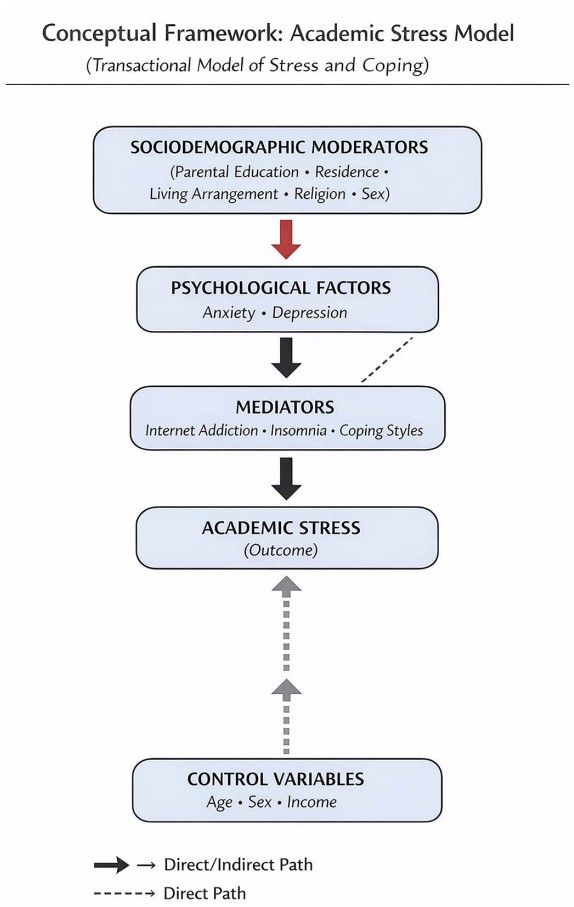

**Fig 1. A Comprehensive Mediated-Moderation Model of Academic Stress.** Note: Solid black arrows indicate hypothesized direct effects; dashed arrows indicate hypothesized indirect/mediated effects. Red arrows denote hypothesized moderating (contextual) effects of sociodemographic variables on the indicated paths (i.e., the strength or direction of the PSY→mediator or PSY→academic stress relationships). Psychological distress (PSY) is hypothesized to exert direct and indirect effects on academic stress via behavioral mediators (sleep quality, self-esteem, internet addiction, coping). Sociodemographic variables are included as exogenous controls and, where indicated (red arrows), as moderators of key paths. Measurement indicators for Psychological distress PSY are depression and anxiety.

Structural Equation Modeling (SEM), enabling simultaneous estimation of direct and indirect pathways within a multivariate system. In addition, network analysis will be conducted to examine the pattern of interrelationships among all observed variables without imposing directional assumptions [Fig 1].

## Method

### Study design, sample size, and sampling

This cross-sectional study was conducted between October 1 and December 30, 2022, among undergraduate medical students enrolled in eight purposively selected public medical colleges in Bangladesh. The institutions were chosen from the country's 37 public medical colleges to ensure geographic representation across all eight administrative divisions, with one medical college included from each divisional headquarters. A stratified random sampling method was used. Students were first divided by academic year (first through fifth year) to ensure proportional representation across all levels of medical education. Within each group, participants were randomly selected using computer-generated random numbers applied to the full class rolls (sampling frame), ensuring every eligible student had an equal chance of being chosen.

The sample size was calculated using these parameters:

Total Population = All medical students across the country (N): 57, 100

Expected Proportion of academic stress among students (p): 54% (0. 54) [18]

Margin of Error (d): ±3% (0. 03)

Confidence Level (Z): 95% (1. 96)

Design Effect (DEFF): 1

Formula

The calculation is based on the formula for a finite population:

$$n = [N \times Z^2 \times p(1 - p)] / [d^2(N - 1) + Z^2 \times p(1 - p)]$$

Calculation Result

N = [57,100 × (1.96)$^2$ × 0.54 × (1−0.54)]/ [(0.03)$^2$ × (57,100−1) + (1.96)$^2$× 0.54 × (1−0.54) = 54,490.75/ 52.34 =1,041.02

Required Sample Size: 1, 042 students.

The initial estimated sample size was 1,042 participants. To account for a potential 5% nonresponse rate, the target sample size was increased to approximately 1,094 students. Each medical college included students from five academic years (first through fifth). From each year, 28 students were randomly selected, resulting in a total of 140 students per college. Across the eight colleges, this sampling method yielded a total of 1,120 selected participants. While all enrolled undergraduates were eligible, those who declined to provide informed consent or were absent due to illness during data collection were excluded. Data were collected using a self-administered structured questionnaire. After screening and cleaning the data to remove incomplete or inconsistent responses, 1,072 fully completed questionnaires were included in the final analysis.

### Ethical statement

Ethical approval for this study was obtained from the Patuakhali Medical College Research Ethics Committee (PkMC-REC-23–05-S1 APPENDIX9). The study was conducted in accordance with the ethical principles of the Declaration of Helsinki. Before participation, all respondents were informed about the objectives and procedures of the study and were assured that participation was entirely voluntary, with the right to withdraw at any time without consequence. Written informed consent was obtained from all participants using a separate consent form. Measures were taken to ensure participants' privacy, confidentiality, and anonymity, and all collected data were used solely for research purposes.

## Data collection and research instrument

Before data collection, administrative approval was obtained through a formal request to the college principal, followed by a briefing session to explain the study procedures to potential participants. Data were collected using a pretested, structured questionnaire administered in Bengali. To ensure linguistic and conceptual equivalence, the instrument was translated and back-translated by bilingual experts. The final questionnaire comprised 132 items across eight sections, capturing psychological, behavioral, and sociodemographic information using validated self-report instruments. These included established measures of academic stress, psychological distress (depression and anxiety), stress coping styles, self-esteem, internet use, and sleep quality, all of which demonstrated satisfactory psychometric properties and had previously been adapted for Bengali-speaking populations where applicable. Sociodemographic variables included age, gender, residence (urban/rural), relationship status, parental education, household income, and religious affiliation. On average, participants required approximately 25 minutes to complete the survey. Data collection for the study took place between mid-October and mid-November 2022, within the broader study window of October 1–December 30, 2022. For context, Bangladesh reported its first COVID-19 case on March 8, 2020; nationwide disruptions to in-person medical education followed, a vaccination campaign began in February 2021, and universities gradually resumed face-to-face teaching from October 2021 under national guidance. Thus, our fieldwork occurred approximately one year after medical colleges reopened. We collected all data in strict accordance with national COVID-19 safety protocols.

**Academic Stress Scale (ASS):** The 40-item Academic Stress Scale, originally developed by Kim (1970) and subsequently adapted for the Indian context by Rajendran and Kaliappan (1990), assesses five domains of academic stress: personal inadequacy, fear of failure, difficulties in interactions with teachers, teacher–pupil relationship and teaching methods, and inadequate study facilities. Each item is rated on a 5-point Likert scale ranging from 0 ("No Stress") to 4 ("Extreme Stress"). The total possible score ranges from 0 to 160, with a cutoff score of 67.13 indicating significant academic stress. The instrument has demonstrated satisfactory psychometric properties, with Cronbach's alpha of 0.70 and test–retest reliability of 0.82 [20,38].

**Stress Coping Style Inventory (SCSI):** The Stress Coping Style Inventory (SCSI) measures four coping dimensions: Active Problem Coping (APC), Active Emotion Coping (AEC), Passive Problem Coping (PPC), and Passive Emotion Coping (PEC). APC directly addresses the stressor through planning and problem-solving, while AEC actively manages emotional responses through strategies such as positive reframing or seeking support. In contrast, PPC reflects avoidance or delay in dealing with the problem, and PEC includes maladaptive emotional responses such as rumination or withdrawal. Active coping strategies generally enhance perceived control and emotional regulation, thereby reducing stress, whereas passive coping tends to maintain or exacerbate stress because the stressor or emotional response remains unresolved. Adapted from Lin & Chen (2010), this 28-item scale assesses four coping styles: active emotional, passive emotional, active problem, and passive problem. Items are rated from 1 ("Completely disagree") to 5 ("Completely agree"), yielding total scores between 28 and 140. The instrument demonstrated excellent reliability, with Cronbach's alpha ranging from 0.86 to 0.88 across subscales, and 0.83 for the overall scale [39].

**Patient Health Questionnaire-9 (PHQ-9):** This 9-item tool assesses depressive symptoms over the previous two weeks, with responses rated on a 0–3 scale. Total scores range from 0 to 27, with standard cut-offs for depression severity (e.g., 5–9 for mild, 10–14 for moderate). The Bengali version used in this study has shown strong internal consistency (Cronbach's $\alpha = 0.84$) and good split-half reliability (0.85) [40].

**Generalized Anxiety Disorder-7 (GAD-7):** A 7-item screening tool for anxiety symptoms, rated from 0 ("Not at all") to 3 ("Nearly every day"). Total scores range from 0 to 21, with a cut-off of 10 indicating clinically significant anxiety. The Bengali adaptation has excellent reliability (Cronbach's $\alpha = 0.90$) and demonstrated good convergent validity in university populations [41].

**Rosenberg Self-Esteem Scale (RSES):** This 10-item scale measures global self-esteem using a 4-point Likert scale (0 = "Strongly Disagree" to 3 = "Strongly Agree"), yielding scores from 0 to 30. Scores below 15 indicate low self-esteem. The scale has high internal consistency (Cronbach's $\alpha = 0.86$) [42].

**Internet Addiction Test (IAT):** A 20-item instrument developed to assess internet addiction severity, covering four domains: lack of control, social withdrawal/emotional conflict, time management issues, and concealment of problematic behavior. Items are rated on a 6-point Likert scale (0 = "Does not apply" to 5 = "Always"), with scores ranging from 0 to 100. The scale demonstrates strong psychometric properties, with a Cronbach's alpha between 0.85 and 0.91, and explains 56.5% of the total variance [43].

**Insomnia Severity Index (ISI):** A 7-item scale assessing insomnia symptoms and their impact, with responses on a 0–4 scale. Total scores range from 0 to 28; scores ≥15 indicate clinically significant insomnia. The ISI has shown excellent reliability (Cronbach's α = 0.90–0.91) and strong convergent validity. A cut-off score of 10 optimally distinguishes insomnia cases with high sensitivity (86.1%) and specificity (87.7%) [44].

## Statistical analysis

Data from this cross-sectional study of 1,072 medical students were screened, cleaned, and analyzed using IBM SPSS Statistics version 23.0 and Jamovi version 2.6.26. Academic stress was dichotomized using a cutoff score of 67 on the Academic Stress Scale, classifying participants as experiencing academic stress or not experiencing academic stress [38].

Descriptive statistics were summarized using frequencies, percentages, and measures of central tendency and dispersion, presented in tables and graphs. Distributional assumptions for continuous variables were assessed before inferential analysis and indicated non-normality. Non-parametric statistical methods were utilized for analysis. The Mann–Whitney U test was conducted to evaluate differences in psychological and behavioral scale scores across different academic stress groups. Additionally, the Chi-square ($\chi^2$) test was done to examine the relationships between categorical variables and academic stress status. All analyses were two-tailed, with results presented including $\chi^2$ statistics, Mann–Whitney U statistics, p-values, and rank-biserial correlation coefficients to estimate effect sizes. Where applicable, crude odds ratios (OR) along with 95% confidence intervals (CI) were calculated.

To identify independent predictors of academic stress, we conducted a binary logistic regression analysis using the Enter method. This method allows all independent variables—including demographic, behavioral, and psychological factors—to be entered into the model simultaneously. By doing so, we can estimate the independent contribution of each predictor while controlling for the effects of the other variables. The model's estimates are reported as regression coefficients (β), adjusted odds ratios (AOR), 95% confidence intervals, and p-values.

To further explore the interplay and structure of relationships among variables, a triangulated analytical strategy was employed, combining logistic regression, structural equation modeling (SEM), and network analysis. SEM was conducted to test a theoretical model in which psychological distress (anxiety and depression) was specified as a latent construct predicting academic stress. Behavioral variables—insomnia, self-esteem, internet addiction, and coping—were modeled as potential mediators, while sociodemographic factors were included as observed covariates. A variety of SEM models were evaluated to determine the best fit based on global fit measures and indices, including the comparative fit index (CFI), root mean square error of approximation (RMSEA), and standardized root mean square residual (SRMR). This included comparisons between the latent-only model and an extended model that incorporated sociodemographic covariates [S1 Appendix], as well as a parsimonious stress-coping style inventory SCSI model versus a segregated stress-coping sub-scales integrated model [S2 Appendix]. To maintain rational statistical reporting standards, we focused on presenting the latent-only model and the parsimonious model of the stress-coping scale inventory (SCSI); however, the extended models are explained in the S1 Appendix, S2 Appendix and Fig 4.

To complement the SEM findings, network analysis was conducted to examine the multivariate interrelationships among psychological distress, behavioral factors, and academic stress within a single system. Centrality metrics—Strength, Closeness, Betweenness, and Expected Influence—were estimated to identify the most influential variables in the network [Fig 5, S3 Appendix and S4 Appendix]. Results from these complementary approaches were integrated into a table to provide a comprehensive understanding of the determinants of academic stress [Table 4].

## Results

The cross-sectional study aimed to find out the prevalence and predictors of academic stress among medical students. Academic stress was absent in 563 participants (52.52%) and present in 509 participants (47.48%). Table 1 reports the distribution of academic stress by sociodemographic factors and associated crude odds ratios (CORs) and χ² tests.
**Sociodemographic correlates [Table 1]:** Religion, living arrangement, residence, dormitory status, monthly expenditure, and year of admission were significantly associated with academic stress. Specifically, participants of religions other than Islam had higher odds of academic stress (COR = 4.21, 95% CI 2.77–6.39, χ² = 51.43, $p < 0.001$). Living in a rural area was associated with increased odds (COR = 1.51, 95% CI 1.06–2.14, χ² = 4.94, $p = 0.026$). Those living with friends had higher odds than those living with family (COR = 2.01, 95% CI 1.53–2.65, χ² = 25.07, $p < 0.001$). Residence in a dormitory

**Table 1. Distribution of Academic Stress Levels by Sociodemographic Factors (N = 1072).**

| Characteristics | Categories | Academic stress | | Total | | | COR with 95% CI |
|---|---|---|---|---|---|---|---|
| | | Absent n (%) | Preset n (%) | N (%) 1072 (100) | χ2(df) | p-value | |
| Sex | Male | 202 (49.4) | 207 (50.6) | 409 (100) | 2.40 (1) | 0.121 | 0.82 (0.64, 1.04) |
| | Female | 361 (54.4) | 302 (45.6) | 663 (100) | | | |
| Age | <=20 years | 245 (54) | 210 (46) | 455 (100) | 0.47 (1) | 0.493 | 1.10 (0.86, 1.40) |
| | >20 years | 318 (52) | 299 (48) | 617 (100) | | | |
| Religion | Islam | 531 (56.7) | 406 (43.3) | 937 (100) | 51.43 (1) | <.0.001* | 4.21 (2.77, 6.39) |
| | Hinduism & others | 32 (23.7) | 103 (76.3) | 135 (100) | | | |
| Mother's Education | =<HSC | 312 (52) | 292 (48) | 604 (100) | 0.41 (1) | 0.520 | 0.92 (0.73, 1.18) |
| | =>Graduate | 251 (54) | 217 (46) | 468 (100) | | | |
| Father's Education | =<HSC | 208 (56) | 162 (44) | 370 (100) | 2.88 (1) | 0.090 | 1.26 (0.97, 1.62) |
| | =>Graduate | 355 (51) | 347 (49) | 702 (100) | | | |
| Personal income | No | 485 (53.5) | 421 (46.5) | 906 (100) | 2.15 (1) | 0.142 | 1.30 (0.93, 1.81) |
| | Yes | 78 (47.0) | 88 (53.0) | 166 (100) | | | |
| Living area | Rural | 93 (61.2) | 59 (38.8) | 152 (100) | 4.94 (1) | 0.026* | 1.51 (1.06, 2.14) |
| | Urban | 470 (51.1) | 450 (48.9) | 920 (100) | | | |
| Living with | Friends | 449 (57.1) | 337 (42.9) | 786 (100) | 25.07 (1) | <0.001* | 2.01 (1.53, 2.65) |
| | Family | 114 (39.9) | 172 (60.1) | 286 (100) | | | |
| Living in a dormitory | No | 8 (15.7) | 43 (84.3) | 51 (100) | 29.13(1) | <0.001* | 6.40 (2.98, 13.75) |
| | Yes | 555 (54.4) | 466 (45.6) | 1021 (100) | | | |
| Marital status | Unmarried | 544 (52.5) | 492 (47.5) | 1036 (100) | .00 (1) | 1.000 | 0.99 (0.51, 1.92) |
| | Married | 19 (52.8) | 17 (47.2) | 36 (100) | | | |
| Monthly Expenditure | Lower | 151 (58) | 108 (42) | 259 (100) | 9.23 (2) | 0.010* | |
| | Middle | 295 (52) | 275 (48) | 570 (100) | | | 1.30 (0.97, 1.75) |
| | Higher | 89 (44) | 113 (56) | 202 (100) | | | 1.78 (1.22, 2.57) |
| Admission session | First year | 134 (61) | 85 (39) | 219 (100) | 49.34 (4) | <0.001* | |
| | Second year | 127 (61) | 82 (39) | 209 (100) | | | 1.02 (0.69, 1.50) |
| | Third year | 133 (61) | 84 (39) | 217 (100) | | | 0.99 (0.68, 1.46) |
| | Fourth year | 89 (43) | 119 (57) | 208 (100) | | | 2.11 (1.43, 3.10) |
| | Fifth year | 80 (37) | 139 (63) | 219 (100) | | | 2.74 (1.86, 4.03) |

* The Chi-square test indicates a statistically significant result at the 0.05 level,

df = Degree of freedom

COR with 95% CI = Crude Odds ratio with 95% Confidence interval

was strongly associated with academic stress (COR = 6.40, 95% CI 2.98–13.75, $\chi^2$ = 29.13, $p < 0.001$). Higher monthly expenditure was associated with greater odds compared with lower expenditure (COR = 1.78, 95% CI 1.22–2.57, $\chi^2$ = 9.23, $p = 0.010$). Fourth- and fifth-year students showed elevated odds relative to first-year students (fourth year COR = 2.11, 95% CI 1.43–3.10; fifth year COR = 2.74, 95% CI 1.86–4.03; overall $\chi^2$ = 49.34, $p < 0.001$). Sex and age were not significantly associated with academic stress in the bivariate tables. [Table 1].

**Psychological and behavioral correlates [Table 2, Fig 2, and Fig 3]:** Table 2 showed graded increases in the prevalence of academic stress with greater anxiety and depression severity (overall $\chi^2$ for anxiety = 72.32, $p < 0.001$; for depression = 179.62, $p < 0.001$). Compared with minimal anxiety, moderate anxiety was associated with higher odds of academic stress (COR = 3.35, 95% CI 2.23–5.02), and severe anxiety also showed elevated odds (COR = 3.56, 95% CI 2.39–5.31). For depression, moderate, moderately severe, and severe categories were associated with progressively larger odds (moderate COR = 4.44, 95% CI 3.03–6.52; moderately severe COR = 6.29, 95% CI 3.95–10.01; severe COR = 26.05, 95% CI 11.86–57.22). Clinical insomnia (moderate–severe) was associated with higher odds of academic stress (COR = 2.41, 95% CI 1.75–3.32, $\chi^2$ = 32.60, $p < 0.001$). Internet addiction showed a strong association: moderate–severe use had greater odds of academic stress (COR = 4.77, 95% CI 3.43–6.63, $\chi^2$ = 95.35, $p < 0.001$). Self-esteem did not differ significantly by stress status (COR = 0.93, 95% CI 0.72–1.22, $p = 0.616$).

Continuous score comparisons (Table 2; Fig 2). Median (IQR) scores were higher among participants with present academic stress for anxiety (median 11 [IQR 9] vs 7 [IQR 7]; Mann–Whitney $p < 0.001$; rank-biserial = 0.31), depression (12 [10] vs 7 [6]; $p < 0.001$; r = 0.43), insomnia (11 [10] vs 8 [8]; $p < 0.001$; r = 0.20), and internet addiction (45 [33] vs 23 [27]; $p < 0.001$; r = 0.37). Self-esteem medians were similar between groups (both 16 [4]; $p = 0.841$).

Fig 2 displayed the distribution (median and IQR) of behavioral scores (anxiety, depression, insomnia, and internet addiction) across academic stress levels and was consistent with the group comparisons reported above. Fig 3 illustrates median scores on academic stress subscales across depression severity categories, showing monotonic increases in subscale medians with greater depression severity.

### Triangulation of logistic regression, SEM, and network analysis

We examined the predictors of academic stress using three complementary approaches: a multivariable logistic regression with a sample size of 1,072 participants [Table 3], a latent structural equation model (SEM) [Fig 4] that utilized weighted least squares mean and variance adjusted (WLSMV) with NLMINB optimization, also involving 1,072 participants. The SEM model converged after 125 iterations and included 26 free parameters. Additionally, we employed a conditional dependence network with centrality analysis [Fig 5]. The model comparison procedures preferred a simpler structural equation model in which stress coping was treated as a single latent construct. Although there is a theoretical distinction between active and passive coping strategies, the overall composite exhibited better fit and structural stability. This supports the decision to use a unitary approach rather than a multi-factor one. Overall fit was adequate, $\chi^2$ (9) = 105.84, $p < .001$, with CFI = 0.96, TLI = 0.90, and SRMR = 0.05. Although comparative indices indicated satisfactory fit, RMSEA remained elevated (0.10, 95% CI 0.08–0.12), suggesting residual misfit [S2 APPENDIX].

**Logistic regression [Table 3]:** All independent variables [variables in Table 1–2] were entered simultaneously into the multivariable logistic regression model to estimate the adjusted effects of the key factors. The logistic regression model showed strong overall performance (Deviance = 1033.53; AIC = 1091.53). Model variance, multicollinearity diagnostics, and classification performance were robust, as detailed in Table 3 [Table 3].

Table 3 presents the multivariable logistic regression results examining sociodemographic, psychological, and behavioral predictors of academic stress. Several sociodemographic factors emerged as significant correlates of the outcome.

Female students had lower odds of academic stress compared to their male counterparts (AOR = 0.54, 95% CI 0.34–0.85, $p = 0.007$). Students aged >20 years also had substantially lower odds of academic stress (AOR = 0.08, 95% CI 0.04–0.17, $p < 0.001$). Compared with first-year students, fourth-year (AOR = 17.08, 95% CI 6.97–41.83, $p < 0.001$) and fifth-year students (AOR = 18.26, 95% CI 6.88–48.50, $p < 0.001$) demonstrated markedly higher odds of academic stress.

**Table 2. Psychological and Behavioral Factors of Academic Stress among Medical Students (N = 1072).**

| Characteristics | Categories | Academic stress | | Total | | | Crude Odds ratio with 95% Confidence interval |
|---|---|---|---|---|---|---|---|
| | | Absent n (%) n = 563 (52.52%) | Preset n (%) n = 509 (47.48) | N (%) 1072 (100) | X²(df) | p-value | |
| **Anxiety** | Minimal | 125 (69.1) | 56 (30.9) | 181 (100) | 72.32 (3) | < 0.001* | |
| | Mild | 236 (62.8) | 140 (37.2) | 376 (100) | | | 1.32 (0.91, 1.93) |
| | Moderate | 98 (40) | 147 (60) | 245 (100) | | | 3.35 (2.23, 5.02) |
| | Severe | 104 (38.5) | 166 (61.5) | 270 (100) | | | 3.56 (2.39, 5.31) |
| **Depression** | None/minimal | 169 (73.8) | 60 (26.2) | 229(100) | 179.62 (4) | < 0.001* | |
| | Mild | 239 (67.5) | 115 (32.5) | 354 (100) | | | 1.36 (0.94, 1.96) |
| | Moderate | 104 (38.8) | 164 (61.2) | 268 (100) | | | 4.44 (3.03, 6.52) |
| | Moderately severe | 43 (30.9) | 96 (69.1) | 139 (100) | | | 6.29 (3.95, 10.01) |
| | Severe | 8(9.8) | 74 (90.2) | 82 (100) | | | 26.05 (11.86, 57.22) |
| **Insomnia** | No clinical insomnia | 239 (59) | 166 (41) | 405 (100) | 32.60 (2) | < 0.001* | |
| | Subthreshold Insomnia | 226 (56) | 179 ('44) | 405(100) | | | 1.14 (.86, 1.51) |
| | Clinical insomnia (Moderate to severe) | 98 (37) | 164 (63) | 262 (100) | | | 2.41 (1.75, 3.32) |
| **Internet addiction** | No significant use | 336 (67) | 168 (33) | 504 (100) | 95.35 (2) | < 0.001* | |
| | Mild internet use | 154 (48) | 167 (52) | 32 (100) | | | 2.17 (1.63, 2.89) |
| | Moderate to severe internet use | 73 (30) | 174 (70) | 247 (100) | | | 4.77 (3.43, 6.63) |
| **Self-esteem** | Low | 157 (51.3) | 149 (48.7) | 306 (100) | 0.25 (1) | 0.616 | 0.93 (0.72, 1.22) |
| | Normal to high | 406 (53) | 360 (47) | 766 (100) | | | |

| | Academic stress | | | | | |
|---|---|---|---|---|---|---|
| | Absent n = 563 (52.52%) | Present n = 509 (47.48) | Non-parametric test for non-normal distribution of scores | | | |
| Total scores of psychological and behavioral factors | Median with Inter-quartile Range (IQR) | Median with Interquartile Range (IQR) | Mann-Whitney U Test statistic | p value | *Rank biserial correlation* | |
| Anxiety (GAD-7) | 7 (7) | 11 (9) | 98213.00 | <0.001[a] | 0.31 | |
| Depression (PHQ-9) | 7 (6) | 12 (10) | 82144.00 | <0.001[a] | 0.43 | |
| Insomnia (ISI) | 8 (8) | 11 (10) | 114102.50 | <0.001[a] | 0.20 | |
| Self-esteem (RSES) | 16 (4) | 16 (4) | 142272.00 | 0.841 | −0.01 | |
| Internet Addiction Test (IAT) | 23 (27) | 45 (33) | 90414.50 | <0.001[a] | 0.37 | |
| Stress Coping Style Inventory | 86 (9) | 88 (12) | 126150.00 | 0.001[a] | 0.12 | |
| Active emotional coping | 30 (6) | 29 (6) | 132432.00 | 0.032[a] | −0.08 | |
| Passive emotional coping | 15 (6) | 16 (5) | 122067.50 | <0.001[a] | 0.15 | |
| Active problem coping | 22 (4) | 22 (6) | 139849.50 | 0.494 | −0.02 | |
| Passive problem coping | 20 (5) | 22 (7) | 122039.50 | <0.001[a] | 0.15 | |

*The Chi-square test indicates a statistically significant result at the 0.05 level

[a] Mann-Whitney U test is significant (p < .05)

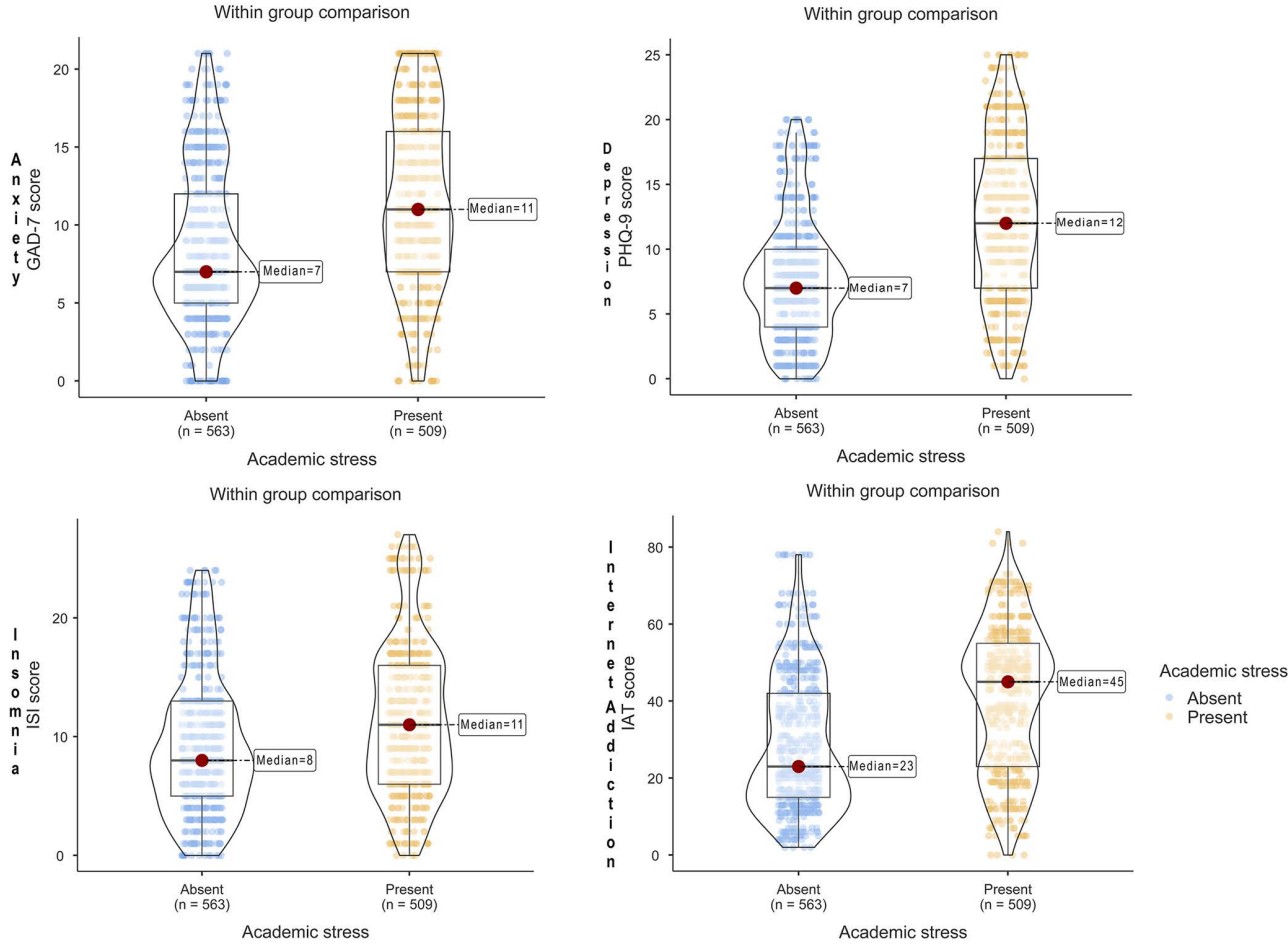

**Fig 2. Distribution of Depression, Anxiety, Insomnia, and Internet Addiction Scores by Academic Stress Status.** Note: Box-violin plots showing the distribution, median, interquartile range, and density of GAD-7 (anxiety), PHQ-9 (depression), ISI (insomnia), and IAT (internet addiction) scores among participants with and without academic stress (Absent vs Present).

Maternal education at graduation level or above was associated with lower odds of academic stress (AOR = 0.26, 95% CI 0.16–0.40, p < 0.001). Students identifying as Hindu or other religions had higher odds of academic stress compared with Muslim students (AOR = 3.16, 95% CI 1.71–5.83, p < 0.001). Living with family was associated with higher odds compared with living with friends (AOR = 1.70, 95% CI 1.08–2.67, p = 0.021), and urban residence was also associated with increased odds (AOR = 2.17, 95% CI 1.22–3.87, p = 0.009). Higher monthly expenditure categories were associated with increased odds of academic stress, particularly the high-expenditure group (AOR = 5.18, 95% CI 2.58–10.40, p < 0.001).

Psychological distress variables demonstrated strong associations with academic stress. Compared with minimal anxiety, mild anxiety (AOR = 2.21, 95% CI 1.23–3.99, p = 0.008) and moderate anxiety (AOR = 3.95, 95% CI 1.98–7.90, p < 0.001) were associated with higher odds of academic stress. Increasing severity of depressive symptoms showed a graded association with the outcome, with severe depression demonstrating the largest effect size (AOR = 21.54, 95% CI 7.21–64.38, p < 0.001).

Among behavioral predictors, sleep disturbance showed significant associations. Subthreshold insomnia was associated with lower odds relative to the reference category (AOR = 0.44, 95% CI 0.28–0.70, p < 0.001), while moderate to severe insomnia showed a stronger association (AOR = 0.26, 95% CI 0.14–0.47, p < 0.001). Internet addiction

 

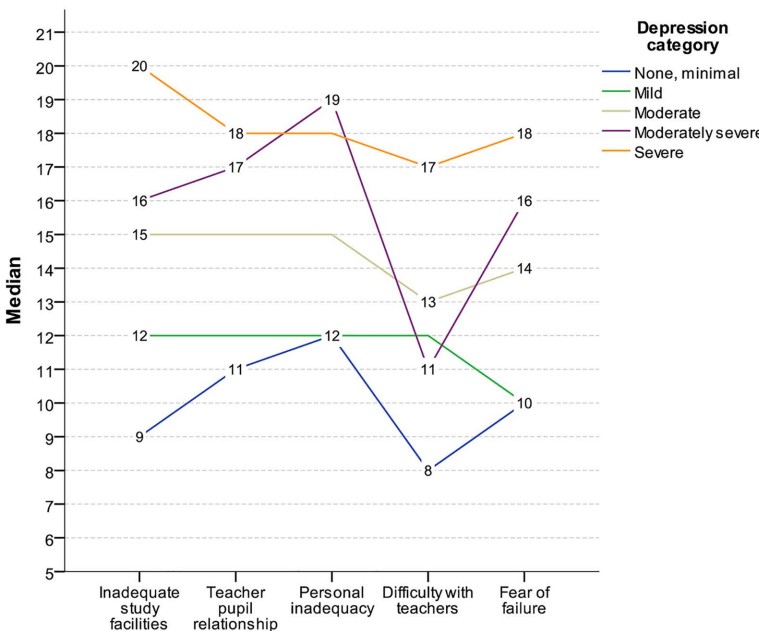

**Academic Stress Sub-Scales Median Scores Across Depression Severity Categories**

**Fig 3. Academic Stress Subscale Medians by Depression Severity.** Note: Median scores for academic stress subscales are plotted across five depression severity categories (none/minimal, mild, moderate, moderately severe, severe).

demonstrated large effect sizes, with mild problematic use associated with increased odds (AOR = 2.17, 95% CI 1.39–3.39, p < 0.001) and moderate-to-severe internet use showing a substantially stronger association (AOR = 17.78, 95% CI 9.66–32.72, p < 0.001).

Among coping strategies, active problem-focused coping showed a statistically significant association with reduced odds of academic stress (AOR = 0.89, 95% CI 0.83–0.95, p < 0.001), whereas other coping domains were not statistically significant.

**Structural Equation Model SEM [Latent-only]:** The SEM corroborated the logistic findings and clarified directional relationships among latent constructs.

Structural regressions showed that PSY exerted a strong direct effect on academic stress (AS_total; β = 1.36, 95% CI 0.72–1.99, p < .001). PSY also predicted sleep quality (SQ_total; β = 0.80, p < .001), self-esteem (SE_total; β = 0.12, p < .001), internet addiction (IAT_total; β = 0.47, p < .001), and stress coping (SC_total; β = 0.22, p < .001).

Indirect effects revealed that PSY increased academic stress via impaired sleep quality (β = −0.65, p = .029) and reduced self-esteem (β = −0.01, p = .030). Indirect pathways through internet addiction and stress coping were non-significant. The total indirect effect was negative (β = −0.67, p = .031), partially offsetting the strong positive direct effect. The overall total effect of PSY on academic stress remained substantial (β = 0.68, 95% CI 0.58–0.78, p < .001).

Explained variance was high for academic stress ($R^2 = 0.72$) and sleep quality ($R^2 = 0.64$), moderate for depression ($R^2 = 0.45$) and anxiety ($R^2 = 0.28$), but low for self-esteem ($R^2 = 0.02$) and stress coping ($R^2 = 0.05$) [Fig 4, S2 Appendix].

**Network structure and centrality:** The conditional dependence network independently identified the same core variables as structurally central [Fig 5, S3 Appendix]. Depression (PHQ) and sleep quality (SQ) exhibited the highest strength and expected influence, with internet addiction (IAT) also among the top central nodes. Self-esteem (SE) and stress-coping (SC) occupied intermediate positions characterized by moderate betweenness, indicating that they bridge psychological distress and behavioral outcomes rather than serving as primary hubs.

**Table 3. Sociodemographic, Psychological, and Behavioral Predictors of Academic Stress (N = 1072).**

| | | Estimate | SE | Z | p | AOR | 95% CI | |
|---|---|---|---|---|---|---|---|---|
| | | | | | | | Lower | Upper |
| Sex | Female – Male* | **−0.62** | **0.23** | **−2.68** | **0.007** | **0.54** | **0.34** | **0.85** |
| Age | >20 – <=20* | **−2.53** | **0.37** | **−6.79** | **<0.001** | **0.08** | **0.04** | **0.17** |
| Admission year | Second year – First year* | 0.37 | 0.31 | 1.17 | 0.243 | 1.44 | 0.78 | 2.67 |
| | Third year – First year* | 0.72 | 0.39 | 1.84 | 0.066 | 2.05 | 0.95 | 4.41 |
| | Fourth year – First year* | **2.84** | **0.46** | **6.21** | **<0.001** | **17.08** | **6.97** | **41.83** |
| | Fifth year – First year* | **2.90** | **0.50** | **5.83** | **<0.001** | **18.26** | **6.88** | **48.50** |
| Father's education | => Graduation – <= Higher secondary* | −0.02 | 0.23 | −0.09 | 0.929 | 0.98 | 0.62 | 1.55 |
| Mother's education | => Graduation – <= Higher secondary* | **−1.36** | **0.23** | **−6.02** | **<0.001** | **0.26** | **0.16** | **0.40** |
| Religion | Hinduism & others – Islam* | **1.15** | **0.31** | **3.67** | **<0.001** | **3.16** | **1.71** | **5.83** |
| Living status | Family – Friends* | **0.53** | **0.23** | **2.31** | **0.021** | **1.70** | **1.08** | **2.67** |
| Living area | Urban – Village* | **0.78** | **0.29** | **2.63** | **0.009** | **2.17** | **1.22** | **3.87** |
| Living place | Home – Hostel* | 0.86 | 0.56 | 1.52 | 0.128 | 2.35 | 0.78 | 7.09 |
| Expenditure category | Middle – Low* | **0.64** | **0.26** | **2.42** | **0.015** | **1.90** | **1.13** | **3.19** |
| | High – Low* | **1.65** | **0.36** | **4.63** | **<0.001** | **5.18** | **2.58** | **10.40** |
| Personal income | Yes – No* | 0.25 | 0.24 | 1.04 | 0.299 | 1.29 | 0.80 | 2.08 |
| Anxiety levels | **Mild – Minimal*** | **0.79** | **0.30** | **2.63** | **0.008** | **2.21** | **1.23** | **3.99** |
| | **Moderate – Minimal*** | **1.37** | **0.35** | **3.90** | **<0.001** | **3.95** | **1.98** | **7.90** |
| | Severe – Minimal* | 0.72 | 0.40 | 1.81 | 0.070 | 2.06 | 0.94 | 4.52 |
| Depression levels | Mild – None/minimal* | −0.49 | 0.27 | −1.78 | 0.075 | 0.62 | 0.36 | 1.05 |
| | Moderate – None/minimal* | **0.88** | **0.32** | **2.78** | **0.005** | **2.42** | **1.30** | **4.51** |
| | Moderately severe –None/minimal* | **1.57** | **0.37** | **4.25** | **<0.001** | **4.80** | **2.33** | **9.89** |
| | Severe – None/minimal* | **3.07** | **0.56** | **5.50** | **<0.001** | **21.54** | **7.21** | **64.38** |
| Self esteem | Normal – Low* | −0.28 | 0.23 | −1.25 | 0.210 | 0.75 | 0.48 | 1.17 |
| Sleep quality | Subthreshold insomnia – No clinically significant insomnia* | **−0.82** | **0.24** | **−3.47** | **<0.001** | **0.44** | **0.28** | **0.70** |
| | Insomnia (moderate to severe) – No clinically significant insomnia* | **−1.36** | **0.31** | **−4.36** | **<0.001** | **0.26** | **0.14** | **0.47** |
| Internet addiction | mild internet use – no significant use* | **0.77** | **0.23** | **3.40** | **<0.001** | **2.17** | **1.39** | **3.39** |
| | moderate to severe internet use – no significant use* | **2.88** | **0.31** | **9.24** | **<0.001** | **17.78** | **9.66** | **32.72** |
| Stress coping | Active emotional coping | −0.03 | 0.02 | −1.25 | 0.210 | 0.97 | 0.93 | 1.02 |
| | Passive emotional coping | −0.04 | 0.03 | −1.40 | 0.161 | 0.97 | 0.92 | 1.01 |
| | Active problem coping | **−0.12** | **0.03** | **−3.64** | **<0.001** | **0.89** | **0.83** | **0.95** |
| | Passive problem coping | −0.03 | 0.03 | −1.06 | 0.291 | 0.97 | 0.93 | 1.02 |

*Note. Estimates represent the log odds of "Academic stress = Present" vs. "Academic stress = Absent"*

*AOR = Adjusted Odds Ratio, 95% CI = 95% confidence interval for Adjusted Odds Ratio (AOR)*

*Model fit measures: Deviance = 1033.53, AIC = 1091.53, McFadden's pseudo-R-squared = 0.32, Nagelkerke's $R^2$ = 0.48.*

*Variance inflation factor VIF range = [1.13 to 2.33] < 4 & Tolerance range = [0.43 to 0.88] >.25.*

*Case classification summary; Accuracy = 0.80, Specificity = 0.83, Sensitivity = 0.78, Area Under Curve AUC = 0.86*

*\* Reference category*

*Bold numbers indicate statistically significant results at the 0.05 level.*

**Synthesis [Table 4]:** The three analytic approaches converged: depressive symptoms, problematic internet use, and sleep disturbance were the most robust correlates of academic stress. SEM indicated that a latent psychological distress factor strongly predicted the behavioral mediators and academic stress, and network centrality measures identified PHQ, SQ, and IAT as the most central nodes. Self-esteem and coping appeared to play secondary or buffering roles

**Fig 4. Structural equation model of psychological distress (latent PSY), behavioral mediators, and academic stress.** Caption: Path diagram showing the latent psychological distress factor PSY (defined by PHQ-9 and GAD-7) and its standardized path coefficients to behavioral mediators (insomnia [SQ_tt], self-esteem [SE_tt], internet addiction [IAT_tt], coping [SC_tt]) and the outcome academic stress (AS_tt). Coefficients displayed are standardized estimates; double-headed arrows indicate covariances. Abbreviations: PSY = psychological distress; PHQ_t = PHQ-9 total score; GAD_t = GAD-7 total score; AS_tt = Academic Stress total score; SE_tt = Self-Esteem total score; SC_tt = Stress Coping Style Inventory total score; SQ_tt = Insomnia Severity Index total score; IAT_tt = Internet Addiction Test total score.

[Table 4]. Given the cross-sectional design, causal claims are tentative; however, the concordance across analytic frameworks strengthens confidence that interventions targeting depression, sleep, and problematic internet use may be effective levers for reducing academic stress.

**Sensitivity and robustness:** We compared a parsimonious (single-coping) SEM and a coping-disaggregated SEM; the parsimonious model provided superior fit (CFI = 0.96; TLI = 0.90; SRMR = 0.05; RMSEA = 0.10, 90% CI [0.08–0.12]) and greater numerical stability. The coping-disaggregated model produced theoretically informative but less stable estimates (lower TLI, higher RMSEA) and evidence of suppression among coping subscales; those results are reported in the Supplement and interpreted cautiously. [S2 Appendix].

## Discussion

This study underscores the complex interplay of psychological, behavioral, and sociodemographic factors contributing to academic stress among medical students. Although consistent with global research, our findings offer distinct perspectives pertinent to the South Asian context, highlighting the diverse origins and critical drivers of academic stress.

### Sociodemographic factors

Our analysis highlights the significant role of sociodemographic factors in predicting academic stress among medical students. Female students were less likely to report stress compared to males (AOR = 0.54, p = .007). This finding diverges from recent studies reporting higher stress among female medical students, often attributed to gendered expectations

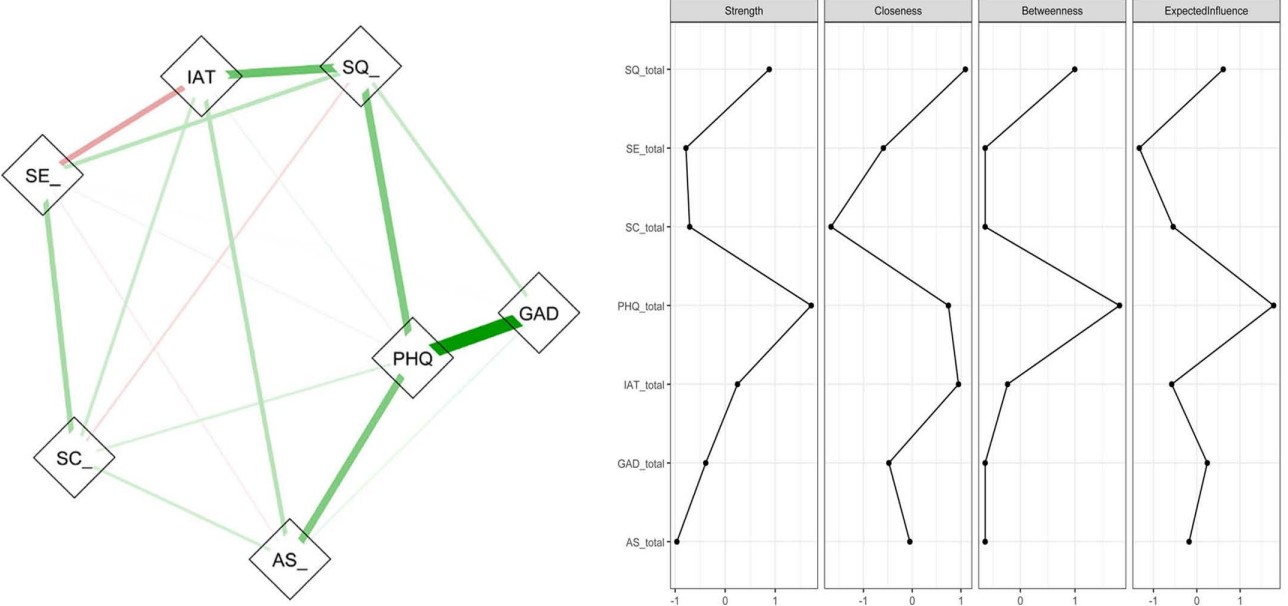

**Fig 5. Network of partial correlations and node centrality for psychological and behavioral correlates of academic stress.** Caption: Left: partial-correlation network among variables (AS_ = Academic Stress; PHQ = PHQ-9 depression; GAD = GAD-7 anxiety; SQ_ = Insomnia Severity Index; IAT = Internet Addiction Test; SE_ = Rosenberg Self-Esteem Scale; SC_ = Stress Coping Style Inventory). Edge color denotes sign (green = positive; red = negative) and edge thickness reflects the magnitude of the partial correlation. Right: standardized centrality indices for each node — Strength, Closeness, Betweenness, and Expected Influence — with higher values indicating greater importance; depressive symptoms, insomnia, and internet addiction are notable central nodes.

and coping differences [45,46], reflecting contextual variations in social support or cultural norms within this study population. Age was strongly protective, with students older than 20 years showing markedly lower odds of stress (AOR = 0.08, p < .001). This aligns with evidence that older students, having developed more mature coping mechanisms and resilience, are less vulnerable to academic stress compared to younger peers who face transitional challenges in adapting to medical education [47,48]. The academic year was a robust predictor, with fourth- and fifth-year students exhibiting substantially higher stress than first-year peers (AORs = 17.08 and 18.26, respectively; p < .001). This pattern is consistent with recent findings that stress escalates across medical training, driven by increased clinical responsibilities, examination pressures, and career uncertainty [17,49]. Expenditure category was positively associated with stress, with students from the middle- and high-expenditure groups showing higher odds than those in the low-expenditure group (AOR = 1.90 and 5.18, respectively). This resonates with research linking higher socioeconomic expectations to academic pressure and performance anxiety, where students from financially stable families may face heightened aspirations and perceived obligations [46,50].

Urban students were approximately twice as likely to report stress as their rural peers, consistent with prior findings linking urban living to heightened academic competition and social pressures [51]. Similarly, students living away from home experienced greater stress, particularly when confronted with language or cultural barriers [4,52]. Living arrangements also played a notable role: those residing with family reported higher stress levels than peers living with friends, a pattern likely attributable to household responsibilities and family expectations. Collectively, these results underscore urban residence and family-related circumstances as the most salient contributors to academic stress in this population. Religious background emerged as an important predictor of stress, with students from minority groups reporting higher levels of distress, consistent with evidence linking minority status and perceived discrimination to adverse mental health outcomes, including academic stress and burnout [53]. Comparable to findings from Pakistan [16], students from

**Table 4. Convergence of findings across methods: logistic regression, latent-only SEM, and network centrality (N = 1,072). [Table 3, Fig 4, Fig 5].**

| Predictor (label) | Logistic regression (adjusted) | Latent-only SEM (standardized path) | Network centrality (relative rank/role) |
|---|---|---|---|
| Depression (PHQ) | Severe vs none: AOR = 21.54 (95% CI 7.21–64.38), p < .001 | PSY → PHQ: β = 0.67 (loading); Estimate = 10.40, SE = 2.90, 95% CI [4.72, 16.08]; z = 3.59, p < .001 | Rank 1 — highest strength and expected influence (network hub) |
| Internet addiction (IAT) | Moderate–severe vs none: AOR = 17.78 (95% CI 9.66–32.72), p < .001 | PSY → IAT: β = 0.47; Estimate = 3.05, SE = 0.26, 95% CI [2.55, 3.56]; z = 11.83, p < .001 | Rank 3 — high strength and expected influence |
| Sleep quality (SQ) | Insomnia (Moderate to severe) vs none: AOR = 0.26 (95% CI 0.14–0.47), p < .001 | PSY → SQ: β = 0.80; Estimate = 1.68, SE = 0.16, 95% CI [1.36, 2.01]; z = 10.27, p < .001. SQ → AS: β = −0.82; Estimate = −2.98, SE = 1.12, 95% CI [−5.18, −0.77]; z = −2.65, p = .008 | Rank 2 — high strength and closeness; central mediator |
| Self-esteem (SE) | Normal vs low: AOR = 0.75 (95% CI 0.48–1.17), p = .210 (ns) | PSY → SE: β = 0.12; Estimate = 0.11, SE = 0.02, 95% CI [0.07, 0.15]; z = 4.98, p < .001. SE → AS: β = −0.10; Estimate = −0.89, SE = 0.29, 95% CI [−1.45, −0.32]; z = −3.09, p = .002 | Rank 4 — moderate centrality; negative expected influence |
| Stress coping (SC) | Active problem coping: AOR = 0.89 (95% CI 0.83–0.95), p < .001; other coping domains non-significant | PSY → SC: β = 0.22; Estimate = 0.55, SE = 0.07, 95% CI [0.41, 0.69]; z = 7.67, p < .001. SC → AS: β = 0.04 (ns) | Rank 5 — moderate betweenness; bridging role between clusters |
| Academic stress (AS) | Outcome (binary) | PSY → AS: β = 1.36; Estimate = 10.40, SE = 2.90, 95% CI [4.72, 16.08]; z = 3.59, p < .001 (latent-only model) | Outcome node in SEM; centrality depends on metric (interpreted via SEM paths) |

Notes. *Logistic AORs and 95% CIs are taken verbatim from the provided logistic table (N = 1,072).*

SEM estimates are standardized βs with corresponding unstandardized estimates, SEs, z, p, and 95% CIs from the latent-only SEM output (WLSMV; bootstrap CIs reported where available).

Network centrality ranks are derived from the latent-only network centrality plots: PHQ and SQ show the highest strength/expected influence, IAT is highly central, SE and SC occupy intermediate positions (bridging roles).

disadvantaged backgrounds—particularly those living at home rather than in dormitories or belonging to minority religious groups—were more vulnerable to stress. In the South Asian context, maternal education has been identified as a protective factor, likely due to enhanced emotional support and academic guidance [54]. Interestingly, paternal education did not show a significant association with stress in our study, which partly contrasts with recent evidence from India highlighting parental expectations and living arrangements as key contributors to medical students' stress [55]. These findings suggest that family dynamics and minority status interact in complex ways to shape stress experiences, underscoring the need for culturally sensitive interventions.

## Psychological and behavioral factors

Psychological distress was strongly associated with academic stress in this study. Students with mild and moderate anxiety demonstrated significantly higher odds of academic stress compared with those reporting minimal anxiety. This finding aligns with previous evidence showing that anxiety is highly prevalent among medical students and is closely linked to academic workload, performance pressure, and frequent evaluations within medical training environments [2,9]. Anxiety can heighten perceived academic demands and impair cognitive functioning, thereby increasing vulnerability to stress.

Depression also showed a strong positive association with academic stress, particularly at moderate to severe levels. Depressive symptoms such as fatigue, reduced motivation, impaired concentration, and negative cognitive appraisal may intensify perceived academic burden and reduce students' ability to cope effectively with demanding curricula [22]. Prior meta-analytic evidence has demonstrated a high prevalence of depression among medical students globally and has consistently linked depressive symptoms with elevated academic stress and psychological distress [8].

Behavioral factors were also important predictors. Problematic internet use showed one of the strongest associations with academic stress, with students experiencing moderate to severe internet addiction having substantially higher odds of stress and depression [56]. Excessive internet use may disrupt time management, reduce academic productivity, and contribute to psychological distress, which in turn may exacerbate academic stress [30]. Recent studies among university students have similarly reported strong associations between internet addiction, poor sleep patterns, and elevated stress levels [57].

Sleep quality demonstrated an inverse association with academic stress in the adjusted model. Although this direction appears counterintuitive, it may reflect a behavioral adaptation common among medical students. Students frequently extend their waking hours to accommodate heavy academic workloads, particularly during examinations or clinical training periods. In this context, reduced sleep duration may function as a short-term coping strategy aimed at completing academic tasks and alleviating perceived academic pressure. While such behavior may temporarily reduce perceived stress by allowing students to meet academic demands, existing literature indicates that persistent sleep deprivation is associated with poorer psychological health and academic performance over time [32,58]. Therefore, this finding may reflect a compensatory coping pattern rather than a beneficial behavioral practice.

Coping strategies further supported this interpretation. Active problem-focused coping was associated with lower odds of academic stress, suggesting that students who actively address academic challenges through planning, task management, and problem-solving may experience better stress regulation. Previous research consistently identifies problem-focused coping as an adaptive strategy that improves psychological resilience and reduces perceived stress in academic settings [24,59].

Self-esteem, however, was not significantly associated with academic stress after adjusting for other psychological variables. This finding suggests that the relationship between self-esteem and stress may operate indirectly through factors such as anxiety, depression, or coping strategies rather than exerting an independent effect.

A triangulated analytical approach integrating logistic regression, structural equation modeling (SEM), and network analysis was employed to identify the principal determinants of academic stress among Bangladeshi medical students. Across all analytical frameworks, depressive symptoms, sleep disturbance, and problematic internet use consistently emerged as the most influential predictors of academic stress. The strong association between depression and academic stress corroborates prior evidence linking psychological distress and internet addiction with persistent symptoms of depression and anxiety among medical students [60], while also highlighting the protective influence of self-esteem [29]. Sleep disturbance likewise appeared as a central factor, aligning with studies demonstrating that poor sleep quality contributes to psychological distress and impaired academic functioning [32], often exacerbated by excessive internet or smartphone use [61]. Problematic internet use further showed a significant behavioral contribution to stress and mental health outcomes [30,60]. In contrast, coping strategies and sociodemographic factors played comparatively limited or moderating roles; however, adaptive coping—particularly active problem-focused coping—and higher self-esteem appeared to function as protective mechanisms that may enhance resilience to academic stress [62].

Collectively, our study findings support the theoretical framework proposed in this study and are consistent with Lazarus and Folkman's Stress and Coping Theory, which posits that psychological distress arises from the dynamic interaction between environmental stressors and individuals' cognitive and behavioral coping resources. From this perspective, behavioral factors such as sleep patterns, internet use, and coping strategies may mediate or modify the relationship between psychological distress and perceived academic stress.

Findings should be interpreted in the context of the COVID-19 pandemic, which the World Health Organization declared a global emergency from March 2020 to May 2023. In Bangladesh, the first case was reported on 8 March 2020, leading to the suspension of in-person medical education and a rapid shift to remote learning. Face-to-face teaching resumed gradually from October 2021 following a national vaccination campaign. This period was marked by disruptions to sleep patterns, lifestyle behaviors, social interaction (with increased reliance on digital platforms), and teaching and assessment

practices. Although data were collected around one year after reopening, students had experienced prolonged academic disruption and social isolation, which may have contributed to persistently elevated academic stress.

## Strengths and limitations

This research has notable strengths, including a large random sample of institutions distributed evenly, which enhances the generalizability of the findings. It used validated psychometric and advanced statistical tools for a meticulous assessment of academic stress and its associated factors. The inclusion of recent literature further adds to its relevance. However, limitations exist. The cross-sectional design restricts causal inferences, and self-reported data may be affected by recall or social desirability bias. Private medical college students were not included in this study, despite their significant presence in the medical student community. Although key confounders were considered, factors like personality traits, institutional or social support, curricular adherence, and resilience were not addressed.

## Conclusion and recommendations

This study underscores that academic stress among medical students is a multifaceted issue with significant implications for medical education, institutional policy, and public mental health. The strongest correlates of academic stress identified include psychological distress (particularly severe depressive and anxiety symptoms), sleep disturbances, and problematic internet use. Additionally, sociodemographic factors such as urban residence, higher paternal education, minority religious affiliation, and specific living arrangements contribute to the vulnerability of students throughout their medical education.

The findings of this study have several practical implications for improving student well-being in medical education. First, routine mental health screening programs could help identify students experiencing anxiety, depression, and academic stress at an early stage, enabling timely support. Second, interventions promoting healthy sleep habits may be beneficial, given the strong association between sleep disturbance and academic stress. Third, strategies aimed at promoting responsible digital behavior and managing excessive internet use may help reduce behavioral risk factors linked to stress. In addition, training programs that strengthen adaptive coping skills—particularly active problem-focused coping—may enhance students' resilience when facing academic demands. Finally, strengthening institutional counseling services and student support systems within medical colleges could provide accessible psychological assistance and structured stress-management programs, thereby contributing to a healthier academic environment.

Future research should employ longitudinal designs to clarify causal pathways and temporal changes in academic stress during medical training. Incorporating additional psychosocial, behavioral, and institutional factors—such as academic burnout, social support, and learning environment—may further improve understanding of the determinants of academic stress. Multi-institutional and cross-cultural studies could help identify contextual differences and inform strategies to enhance student wellbeing. In addition, rigorous evaluations of existing intervention programs are needed to assess their effectiveness and sustainability in reducing academic stress and improving the learning environment for medical students.

## Supporting information

**S1 Appendix. Structural Equation Modeling of Psychological Distress, Behavioral Mediators, and Academic Stress with Sociodemographic Adjustments.**
(PDF)

**S2 Appendix. Comparison of Stress Coping Styles: Segregated vs. Parsimonious Structural Equation Modeling (SEM).**
(PDF)

**S3 Appendix. Latent-only variables network and centrality graph.** [Sociodemographic variables not included]. (PDF)

**S4 Appendix. Network and centrality analysis of psychological, behavioral, and sociodemographic variables related to academic stress.** (PDF)

## Acknowledgments

Department of Community Medicine and Public Health of Patuakhali Medical College, Directorate General of Medical Education.

## Author contributions

**Conceptualization:** Md Rizwanul Karim, Faiza Rumeen, Purna Aruneema.

**Data curation:** Md Rizwanul Karim, S. A. Sazin Haque, Faiza Rumeen.

**Formal analysis:** Md Rizwanul Karim, Purna Aruneema.

**Investigation:** S. A. Sazin Haque.

**Methodology:** Md Rizwanul Karim, S. A. Sazin Haque, Faiza Rumeen.

**Software:** Md Rizwanul Karim, Faiza Rumeen.

**Supervision:** S. A. Sazin Haque, Faiza Rumeen, Purna Aruneema.

**Validation:** Md Rizwanul Karim.

**Writing – original draft:** Md Rizwanul Karim.

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
