## [Decision Letter · Decision Letter 0]

18 Feb 2026

PONE-D-25-26733Academic Stress and Its Psychosocial and Behavioral Determinants in Medical Students: Findings from a Cross-Sectional StudyPLOS One

Dear Dr. Karim,

Thank you for submitting your manuscript to PLOS ONE. After careful consideration, we feel that it has merit but does not fully meet PLOS ONE’s publication criteria as it currently stands. Therefore, we invite you to submit a revised version of the manuscript that addresses the points raised during the review process.

We look forward to receiving your revised manuscript.

Kind regards,

Dr. Tailson Evangelista Mariano

Academic Editor

PLOS One

Journal Requirements:

“This study was undertaken as part of the academic curriculum for 4th-phase medical students during the Residential Field Site Training (RFST) program. It was conducted independently, without financial assistance from public, private, commercial, or non-profit organizations. Moreover, the authors did not receive any remuneration or honorarium for their involvement in the research.”

“This study was undertaken as part of the academic curriculum for 4th-phase medical students during the Residential Field Site Training (RFST) program. It was conducted independently, without financial assistance from public, private, commercial, or non-profit organizations. Moreover, the authors did not receive any remuneration or honorarium for their involvement in the research.”

**Additional Editor Comments:**

Line: 126: Remove the link and do proper Citation

Under the results chapter need to close the results finding with brackets

Reviewers' comments:

Reviewer's Responses to Questions

**Comments to the Author**

1. Is the manuscript technically sound, and do the data support the conclusions?

Reviewer #1: Partly

Reviewer #2: Yes

2. Has the statistical analysis been performed appropriately and rigorously?

Reviewer #1: No

Reviewer #2: Yes

3. Have the authors made all data underlying the findings in their manuscript fully available?

The PLOS Data policy requires authors to make all data underlying the findings described in their manuscript fully available without restriction, with rare exception (please refer to the Data Availability Statement in the manuscript PDF file). The data should be provided as part of the manuscript or its supporting information, or deposited to a public repository. For example, in addition to summary statistics, the data points behind means, medians and variance measures should be available. If there are restrictions on publicly sharing data—e.g. participant privacy or use of data from a third party—those must be specified. requires authors to make all data underlying the findings described in their manuscript fully available without restriction, with rare exception (please refer to the Data Availability Statement in the manuscript PDF file). The data should be provided as part of the manuscript or its supporting information, or deposited to a public repository. For example, in addition to summary statistics, the data points behind means, medians and variance measures should be available. If there are restrictions on publicly sharing data—e.g. participant privacy or use of data from a third party—those must be specified. requires authors to make all data underlying the findings described in their manuscript fully available without restriction, with rare exception (please refer to the Data Availability Statement in the manuscript PDF file). The data should be provided as part of the manuscript or its supporting information, or deposited to a public repository. For example, in addition to summary statistics, the data points behind means, medians and variance measures should be available. If there are restrictions on publicly sharing data—e.g. participant privacy or use of data from a third party—those must be specified. requires authors to make all data underlying the findings described in their manuscript fully available without restriction, with rare exception (please refer to the Data Availability Statement in the manuscript PDF file). The data should be provided as part of the manuscript or its supporting information, or deposited to a public repository. For example, in addition to summary statistics, the data points behind means, medians and variance measures should be available. If there are restrictions on publicly sharing data—e.g. participant privacy or use of data from a third party—those must be specified.

Reviewer #1: Yes

Reviewer #2: Yes

4. Is the manuscript presented in an intelligible fashion and written in standard English?

Reviewer #1: No

Reviewer #2: Yes

5. Review Comments to the Author

Reviewer #1: The manuscript addresses a relevant topic and presents two major strengths: an adequate sample size and an interesting set of variables. Considering these aspects, I outline below the main suggestions aimed at achieving a version suitable for publication.

1. Introduction

1.1. Include data on Bangladesh with regard to the variables investigated in the study.

1.2. The timing of data collection coincides with the COVID-19 pandemic. This aspect needs to be considered in the introduction and, above all, in the discussion of the manuscript.

1.3. Some issues hinder the psychological interpretation of the data. For example, regarding the Stress Coping Style Inventory, it is important to briefly explain what each strategy means (i.e., coping styles: active emotional, passive emotional, active problem, and passive problem coping). The key question is: are all these strategies dysfunctional? Without prior clarification before the discussion section, readers may be led to inaccurate conclusions.

2. Methods

2.1. Please see suggestions 5.4.1, 5.4.2, and 5.5.

3. Results

3.1. In the logistic regression analysis, were all variables entered simultaneously?

3.2. Please clarify whether control variables were used in the analyses. If so, indicate when they were included and how they were handled. There is mention of control variables in the text, but this is not clearly specified.

3.3. Please see suggestions 5.4.1, 5.4.2, and 5.5.

4. Discussion

4.4. The discussion section of the manuscript is appropriate. However, based on the suggestions outlined below, modifications will be necessary.

4.5. Even prior to addressing the suggestions, it is important that the authors employ one or more theoretical frameworks (or previous empirical evidence) to explain the relationships among the variables.

5. Major Suggestions

5.1. Orthographic and technical revision of the manuscript: apply more technical and objective language.

5.1.1. Example 1: “Researchers created a forest plot to illustrate the primary predictors of academic stress using the results from a logistic regression model.” A more appropriate wording is: “The predictors of academic stress were illustrated using a forest plot based on the results of the logistic regression analysis.”

5.1.2. Example 2: “This mirrors findings from a recent Indian study that identified parental expectations and living arrangements as major contributors to stress among medical students (30)”. This excerpt could be written using structures such as: “These results are consistent with findings reported in a recent study…”; “Our observations align with those of a recent Indian study…”; “Similar findings have been documented in a recent study…”.

5.1.3. Example 3: in the excerpt “Insomnia was significantly associated with academic stress, echoing findings that poor sleep impairs cognitive function, emotional regulation, and stress resilience (32).”, the expression “echoing findings” is somewhat uncommon in technical writing for empirical articles.

5.2. The figures need to be improved in terms of quality.

5.3. The tables appear to have formatting issues in their lower sections.

5.4. The authors enumerate the strengths of the manuscript (e.g., large random sample, validated psychometric tools and advanced multivariate logistic regression). However, from a technical standpoint, the analytical aspect is imprecise. More specifically, given the number of variables (i.e., the factors of each measure), the analytical procedure adopted is overly simplistic. Therefore, two plausible suggestions are presented:

5.4.1. Use Structural Equation Modeling (SEM) to test an explanatory model. One suggestion is to test mediators and moderators to explain academic stress (dependent variable). In this framework, psychological aspects (anxiety and depression) would be the independent variables. Sociodemographic variables could function as control and/or moderating/mediating variables. Behavioral variables could likewise serve this role (i.e., as control and/or moderating/mediating variables). In this case, the role of sociodemographic and behavioral variables in the relationship between psychological aspects and academic stress would be tested. Consequently, theory and/or prior evidence must be used to justify the hypotheses and the function assigned to sociodemographic and behavioral variables.

5.4.2. Use Network Analysis. This technique is a powerful tool for understanding, visualizing, and quantifying patterns of interdependence. Importantly, all variables would be considered within a single analytical procedure (i.e., truly multivariate). This approach would allow for the evaluation of Betweenness (measures the extent to which a variable acts as a bridge between other variables in the network), Closeness (measures how close a variable is to all other variables in the network), Strength (measures the strength or number of connections a variable has with others), and Expected Influence (measures the overall impact of a variable on the network, considering both the direction - positive or negative - and the strength of the connections). Consequently, network analysis would enable the visualization of complex relationships and the prioritization of key variables for intervention, offering a richer understanding of academic stress than traditional analyses.

5.5. The authors state that “This study highlights the complex interplay of factors…”. For this claim to be truly plausible, it is necessary to conduct appropriate analytical procedures (see points 5.4.1 and 5.4.2).

5.6. There is no section on practical implications in the manuscript. A standardized section is needed to concretely explore possible intervention pathways, rather than offering vague and generic mentions.

Reviewer #2: Dear Authors,

First and foremost, I would like to commend you on this robust, timely, and well-conducted study. Investigating academic stress among medical students is of paramount importance, as these individuals will become future professionals entrusted with human lives. The mental, emotional, and physical well-being of medical students inevitably influences the quality of care they will provide to the broader population. In this sense, your work addresses a highly relevant public health and educational concern.

Below, I offer some suggestions to further strengthen the manuscript:

Introduction:

While the gap in the literature is clearly articulated, I encourage you to expand the justification for the study by more explicitly highlighting its social and methodological contributions. Beyond addressing an academic gap, what are the broader societal implications of your findings? How might they inform institutional policies, mental health interventions, or educational reforms? Additionally, clarifying the methodological contributions of the study (such as the integration of multiple predictors within a single analytical framework) would further reinforce the originality and added value of the research. Strengthening these aspects would enhance the overall rationale and impact of the manuscript.

Methods:

The study benefits from a large and well-distributed sample, which significantly strengthens the robustness of the analyses and the generalizability of the findings. This is an important methodological strength and should be clearly emphasized as such in the manuscript.

Results:

The results are presented in a clear, coherent, and methodologically appropriate manner. No additional comments or revisions are necessary.

Discussion:

No additional comments. I consider this section to be very well written, coherent, and appropriately supported by the literature.

Conclusion and Recommendations:

I recommend strengthening this section by emphasizing the broader scientific, educational, and social significance of the study. Rather than reiterating previously presented results at the outset (which makes the section somewhat repetitive) it would be more impactful to focus on the implications of the findings for medical education, mental health policy, and institutional practices.

If the authors consider it essential to retain a concise summary of the main findings, I suggest relocating this synthesis to the end of the Discussion section, where it would serve as a natural transition into the conclusion. This adjustment would improve the manuscript’s structure, reduce redundancy, and enhance the overall coherence and flow of the paper.

It would be valuable to include explicit recommendations for future research based on the limitations identified in the present study. Doing so would not only strengthen the scientific contribution of the manuscript but also provide meaningful guidance for future researchers in this field. Clearly outlining potential directions (such as longitudinal designs, inclusion of additional psychosocial variables, or cross-cultural comparisons) would enhance the article’s impact and relevance for advancing the literature on academic stress.

General recommendations: It is strongly recommended that the manuscript undergo a careful and comprehensive proofreading process to address spelling, grammatical, and stylistic issues. Additionally, the authors should thoroughly review the journal’s submission guidelines to ensure full adherence to all editorial and formatting requirements.

6. PLOS authors have the option to publish the peer review history of their article (what does this mean?). If published, this will include your full peer review and any attached files.). If published, this will include your full peer review and any attached files.). If published, this will include your full peer review and any attached files.). If published, this will include your full peer review and any attached files.

**Do you want your identity to be public for this peer review?** For information about this choice, including consent withdrawal, please see our  For information about this choice, including consent withdrawal, please see our  For information about this choice, including consent withdrawal, please see our  For information about this choice, including consent withdrawal, please see our Privacy Policy....

Reviewer #1: No

Reviewer #2: No

---

## [Author Response · Author response to Decision Letter 1]

26 Mar 2026

We sincerely thank the Editor and Reviewers for their careful evaluation and constructive feedback, which have been invaluable in strengthening our manuscript. Their insightful comments guided a comprehensive and systematic revision of the work. In response, we have submitted both a tracked-changes version and a clean version of the manuscript, and have implemented substantial revisions across all sections, from the Introduction through to the Conclusion.

The Introduction has been refined to incorporate relevant evidence from Bangladesh on academic stress, thereby more clearly identifying the existing research gap. It now more effectively highlights the societal relevance and methodological contributions of the study. The Conclusion has also been revised to emphasize the novelty of our findings, their broader implications, and directions for future research. To strengthen the conceptual underpinning, we have integrated Lazarus and Folkman’s stress and coping model into the manuscript. In addition, we have provided a detailed account of the sample size determination process and improved the presentation of results. Specifically, the graphical displays have been revised by replacing violin plots and line charts with median and interquartile range–based summaries, ensuring appropriate representation of the non-normal distribution of the data.

Furthermore, in line with the reviewers’ recommendations, we have adopted more advanced analytical approaches, including Structural Equation Modeling (SEM) and network analysis, and have introduced corresponding network and centrality figures. The reference list has been carefully updated and expanded to reflect the most relevant and recent literature. Additional supplementary materials have been included to enhance methodological transparency, including the rationale for employing a parsimonious latent-variable SEM model, justification for excluding extended sociodemographic covariates, and detailed reporting of model fit indices. A summary table has also been added to clearly demonstrate the triangulation of findings across the different analytical approaches.

Collectively, these revisions have resulted in a more coherent, methodologically rigorous, and substantially strengthened manuscript. We are deeply grateful for the reviewers’ constructive suggestions, which have significantly improved the clarity, rigor, and overall quality of our work. Below, we provide a detailed, point-by-point response to all comments

Journal Requirement 1

Comment: Ensure manuscript meets PLOS ONE formatting and style requirements.

Response: We have revised the manuscript to fully comply with PLOS ONE formatting guidelines, including file naming, structure, and style templates.

Journal Requirement 2

Comment: Mismatch between Funding Information and Financial Disclosure.

Response: We have carefully reviewed and corrected inconsistencies between the Funding Information and Financial Disclosure sections.

“Financial Disclosure: This study was conducted as part of the academic curriculum for 4th-phase medical students during the Residential Field Site Training (RFST) program. The study was carried out independently and did not receive any specific grant from funding agencies in the public, commercial, or not-for-profit sectors. No additional external funding was received for this study. “[Line 696-700, p 34].

Journal Requirement 3

Comment: Amend the funding statement and include the required sentence “Please also include the statement, 'There was no additional external funding received for this study.’ in your updated Funding Statement.

Response: The funding statement has been revised to clearly declare all sources of support. We have also included the required sentence: “No additional external funding was received for this study.” [Line 700, p 34].

Journal Requirement 4

Comment: Clarify the role of funders.

Response: We confirm that no funders were involved. The statement has been added:

“Role of Funders: The funders had no role in study design, data collection and analysis, decision to publish, or preparation of the manuscript.” [Line 701-3, p 34].

Journal Requirement 5

Comment: Ethics statement placement.

Response: The ethics statement has been moved exclusively to the Methods section and removed from all other sections [Line 211-19, p 10].

Journal Requirement 6

Comment: Include captions for Supporting Information.

Response: Captions for all supporting information files have been added and cross-referenced appropriately in the manuscript.

Editor Comment

Comment: Line 126: Remove link and use proper citation.

Response: The hyperlink has been removed and substituted with the standard formula for determining sample size in cross-sectional studies, along with calculations [Line 189-201, p 9].

Editor Comment

Comment: Close results findings with brackets.

Response: All results sections have been revised to ensure proper formatting and closure of statistical reporting.

Reviewer 1

General remark: “The manuscript addresses a relevant topic and presents two major strengths: an adequate sample size and an interesting set of variables. Considering these aspects, I outline below the main suggestions aimed at achieving a version suitable for publication.”

General response: Thank you very much for recognizing the relevance of our topic and highlighting the strengths of the manuscript, particularly the adequate sample size and the diverse set of variables. We sincerely appreciate your constructive feedback and detailed suggestions, which have guided us in refining the analytical framework, strengthening methodological rigor, and enhancing the clarity of our revision. Your comments have been invaluable in improving the overall quality and publication readiness of the manuscript.

Comment 1.1 — Introduction: Include data on Bangladesh about the variables investigated in the study.

Response: We have expanded the Introduction to include additional, Bangladesh specific evidence on prevalence and correlates of academic stress, depression, anxiety, sleep disturbance, and internet use among tertiary and medical students. We cite national studies and briefly summarize their findings to situate our work in the local context.

Revision made: Introduction — added one paragraph summarizing prior Bangladeshi studies and their findings (new text added to the end of the original Introduction). These paragraphs explicitly reference the multicenter and single site studies cited in the original manuscript and clarify how our study extends that evidence. [line 100-109, p 5]

Comment 1.2 — Introduction/Discussion: Data collection coincided with the COVID 19 pandemic; this needs to be considered.

Response: We acknowledge the timing of data collection and its potential influence on psychological and behavioral measures. We added a dedicated paragraph in the Methods describing the data collection period (October–December 2022). In the Discussion, we explicitly consider how pandemic-related disruptions and residual effects (e.g., altered learning formats, social restrictions) may have influenced prevalence estimates and behavioral patterns.

We also discuss the direction and magnitude of potential bias and note that the multicenter sampling and validated instruments mitigate, but do not eliminate, this concern.

Revision made: Introduction- Covid-19 effect on academic stress [Line 110-12, p 5]; Methods (Data collection subsection) — clarified dates and contextualized them with respect to the pandemic [Line 234-39, p 11]; Discussion — added explanation discussing possible pandemic effects and how they were considered in interpretation [Line 647-55, p 31-32].

Comment 1.3 — Clarify Stress Coping Style Inventory subscales and whether strategies are dysfunctional.

Response: We added a concise description of each SCSI subscale (Active Problem Coping, Active Emotion Coping, Passive Problem Coping, Passive Emotion Coping), clarifying which strategies are generally considered adaptive versus maladaptive in the literature and how we interpret higher scores on each subscale in relation to academic stress. This clarification prevents misinterpretation of the coping results in the Discussion.

Revision made: Methods (Measures subsection: Stress Coping Style Inventory) — added description defining each subscale and indicating typical adaptive/maladaptive classification [line 248-56, p 11-12]; Discussion — referenced these definitions when interpreting coping results [line 616-21, p 30].

Comment 2.1 — Methods: See suggestions 5.4.1, 5.4.2, and 5.5 (analytical precision).

Response: We implemented the recommended advanced analyses to address the complexity of relationships among variables: (a) Structural Equation Modeling (SEM) to test the hypothesized mediated pathways with psychological distress as a latent construct and behavioral variables as mediators; and (b) Network Analysis to visualize and quantify interdependencies and centrality of variables. Both analyses were added to the manuscript, and their results were integrated with the logistic regression findings. These additions directly address the reviewer’s concern about overly simplistic analysis.

Multiple SEM models were evaluated to determine the best fit based on global fit measures and indices. This included comparisons between the latent-only model and an extended model that incorporated sociodemographic covariates, as well as a parsimonious stress-coping model versus a segregated stress-coping sub-scales integrated model. To maintain rigorous statistical reporting standards, we focused on presenting the latent-only model and the stress-coping scale inventory (SCSI) parsimonious model; however, the extended models are included in the supplementary files.

Revision made: Methods (Statistical analysis subsection) — expanded to describe SEM specification (latent PSY indicated by PHQ 9 and GAD 7; mediators: ISI (sleep quality, SQ); IAT (internet addiction); RSES (self-esteem); SCSI (stress coping style inventory) subscales; sociodemographic covariates included as exogenous controls) and network analysis procedures (estimation method, centrality metrics) [Line 307-27, p 14-15] Supplementary Materials (S1, S2, S3, S4).

Results — added SEM fit indices and path coefficients; added network graph and centrality table (new Figure and Table). Supplementary Materials (S1, S2, S3, S4) — provided full SEM diagrams and network plots [Line 406-539, p 20-26].

Comment 3.1 and 3.2 — Results: In logistic regression, were all variables entered simultaneously? Clarify control variables.

Response: Yes. We clarified that the binary logistic regression used the Enter method, with all candidate predictors entered simultaneously to estimate adjusted effects. We explicitly list the control variables (sex, age, residence, family type, parental education, monthly expenditure, and academic year) and indicate how categorical variables were dummy coded. We also report multicollinearity diagnostics (VIF) and model fit statistics.

Revision made: Methods (Statistical analysis subsection) — added explicit statement that the Enter method was used and listed control variables and coding; [Line 301-3, p 14].

Results (Regression subsection) — added VIF results and model fit information; Table — updated to show adjusted odds ratios (AOR), 95% CIs, p values, and footnotes explaining the model fit [Line 420-24, p 20-21; Table 3, p 23].

Comment 4.5 — Use theory and prior evidence to justify SEM and moderator/mediator roles.

Response: We expanded the Theoretical Framework section to more explicitly link the Transactional Model of Stress and Coping to our SEM specification. We justify the selection of mediators (sleep disturbance, internet addiction, self esteem, coping) and the inclusion of sociodemographic variables as exogenous controls and potential moderators, citing prior empirical studies that support these roles. Moderator tests were performed where theoretically justified (e.g., testing whether residence or academic year moderated the PSY → academic stress path). Results of the moderation tests are reported.

Revision made: Theoretical Framework — added explicit rationale and citations for mediator/moderator assignments [142-69, p 7-8]; Methods (SEM subsection) — described moderator tests [Line 307-9, p 14]; Results — reported moderation test outcomes [Line 478-89, p 24; Line 516-23, p 25, Figure 1, Figure 4, Figure 5, Table 4, p-26].

Comment 5.1 — Language and style: Orthographic and technical revision of the manuscript: apply more technical and objective language.

Response: We performed a comprehensive language edit to improve technical tone and clarity. Examples provided by the reviewer were rephrased using more objective scientific language throughout the manuscript.

5.1.1- Forest plot removed to avoid duplicating the logistic regression findings. Other important figures, such as Figure 4, Figure 5, and supplementary materials (S1, S2, S3, S4), illustrate the findings with clarity and precision.

5.1.2- revised as recommended [Line 556-63, p 27]

5.1.3- revised a recommended [Line 588-96, p 29]

Revision made: Full manuscript revised for language and style; specific sentences reworded as suggested (see tracked changes in the submitted revision).

Comment 5.2 and 5.3 — Figures and tables quality/formatting.

Response: We improved figure resolution and formatting, standardized fonts and labels, and corrected table formatting issues in the lower sections. All figures were regenerated for clarity; tables were reformatted to ensure consistent alignment and legibility.

Revision made: Figures 1–5 and Tables 1–4 — replaced with higher resolution versions and corrected formatting; figure captions expanded for clarity.

Comment 5.4 [5.5.1, 5.4.2] & 5.5 — We appreciate the reviewer’s constructive observation regarding the analytical precision of our study. In line with the suggestions, we have substantially strengthened the analytical framework:

Response: Structural Equation Modeling (SEM): We implemented SEM to test an explanatory model guided by the Transactional Model of Stress and Coping. Psychological distress (latent construct indicated by depression and anxiety) was modeled as the independent variable, academic stress as the dependent variable, and behavioral factors (sleep disturbance, internet addiction, self esteem, coping styles) as mediators. Sociodemographic variables were incorporated as exogenous controls and, where theoretically justified, as moderators. Fit indices (CFI, RMSEA, SRMR) and indirect effects (bootstrap estimates) are reported, allowing us to rigorously assess both direct and indirect pathways.

Two Lavaan syntaxes were used

Latent-only SEM model in lavaan model <- '

# 1. Measurement Model

PSY =~ GAD_total + PHQ_total

# 2. Structural Model

# Direct effect of PSY on academic stress

AS_total ~ c*PSY

# Mediator regressions

SQ_total ~ a1*PSY

SE_total ~ a2*PSY

IAT_total ~ a3*PSY

SC_total ~ a4*PSY

# Academic stress predicted by mediators

AS_total ~ b1*SQ_total + b2*SE_total + b3*IAT_total + b4*SC_total

# 3. Indirect Effects

ind_SQ := a1*b1

ind_SE := a2*b2

ind_IAT := a3*b3

ind_SC := a4*b4

total_ind := ind_SQ + ind_SE + ind_IAT + ind_SC

# 4. Measuring covariances

GAD_total ~~ PHQ_total

# 5. Total Effect

total := c + total_ind

Extended Measurement Model including sociodemographic controls

# Latent Psychological Distress

PSY =~ GAD_total + PHQ_total

# 2. Structural Model

# Direct effect

AS_total ~ c*PSY

# Mediator regressions

SQ_total ~ a1*PSY

SE_total ~ a2*PSY

IAT_total ~ a3*PSY

SC_total ~ a4*PSY

# Academic stress predicted by mediators

AS_total ~ b1*SQ_total + b2*SE_total +

b3*IAT_total + b4*SC_total

# 3. Sociodemographic Controls

AS_total ~ Live_area_2cat + Mother_edu_2 + Famtype_2cat +

Expen_cat + Sex

SQ_total ~ Ad_yr + Sex

SE_total ~ Ad_yr + Sex

IAT_total ~ Ad_yr + Sex

SC_total ~ Ad_yr + Sex

# 4. Indirect Effects

ind_SQ := a1*b1

ind_SE := a2*b2

ind_IAT := a3*b3

ind_SC := a4*b4

total_ind := ind_SQ + ind_SE + ind_IAT + ind_SC

GAD_total ~~ PHQ_total

# 5. Total Effect

total := c + total_ind

[Line 407-19, p 20; Line 476-89, p 24]; Figure 4; S1 & S2.

---

## [Editor Report · Decision Letter 1]

1 Apr 2026

Academic Stress and Its Psychosocial and Behavioral Determinants in Medical Students: Findings from a Cross-Sectional Study

PONE-D-25-26733R1

Dear Dr. Karim,

We’re pleased to inform you that your manuscript has been judged scientifically suitable for publication and will be formally accepted for publication once it meets all outstanding technical requirements.

Kind regards,

Tailson Evangelista Mariano, Ph.D.

Academic Editor

PLOS One
---

## [Editor Report · Acceptance letter]

PONE-D-25-26733R1

PLOS One

Dear Dr. Karim,

I'm pleased to inform you that your manuscript has been deemed suitable for publication in PLOS One. Congratulations! Your manuscript is now being handed over to our production team.

Kind regards,

on behalf of

Dr. Tailson Evangelista Mariano

Academic Editor

PLOS One